# When a Pandemic Strikes: Resilience of Swedish Academics in the Face of Coronavirus

**DOI:** 10.3390/ijerph192013346

**Published:** 2022-10-16

**Authors:** Önver Andreas Cetrez, Saeid Zandi, Fereshteh Ahmadi

**Affiliations:** 1Department of Psychology of Religion, Faculty of Theology, Uppsala University, Box 511, 75120 Uppsala, Sweden; 2Department of Social Work and Criminology, Faculty of Health and Occupational Studies, University of Gävle, 80176 Gävle, Sweden

**Keywords:** coping methods, COVID-19, crisis, culture, epidemic, higher education, resiliency, Sweden, university community, university staff

## Abstract

Background: The COVID-19 pandemic hit the world with severe health consequences, affecting some populations more than others. One understudied population is the academic community. This study, part of a larger project looking at COVID-19 in Sweden and internationally, aims to understand the individual and collective dimensions of resilience among academics in Sweden during the early wave of the pandemic. Method: A quantitative research design was applied for this cross-sectional study. We used simple random sampling, administered through an online survey, on academics at Swedish universities (*n* = 278, 64% women). We employed the CD-RISC 2 (the Connor–Davidson Resilience Scale) to measure personal/individual resilience, additional items for social/collective resilience, and a meaning-making coping instrument (meaning, control, comfort/spirituality, intimacy/spirituality, life transformation). Results: The results revealed a strong level of personal/individual resilience among men (*M* = 6.05) and a level just below strong among women (*M* = 5.90). By age group, those 35–49-year-olds showed strong resilience (*M* = 6.31). Family was the dominant social/collective resilience factor, followed by friends, nature, work/school, and, lastly, religion/spirituality. There was a positive and significant correlation between self-rated health and personal/individual resilience (*r* = 0.252, *p* = 0.001) and positive but weak correlations and negative significant correlations between personal/individual resilience and religious coping methods. Conclusions: During the pandemic, the family took priority in meaning-making, which is an interesting change in a strong individual-oriented society such as Sweden.

## 1. Introduction

### 1.1. Previous Research

In March 2020, COVID-19 was announced as a pandemic by the WHO, and Sweden followed soon after. The rise and intensification of depression, suicide, substance use, and domestic violence are some effects of the COVID-19 pandemic. Researchers have warned against the COVID-19 pandemic’s impact on psychological health [1,2], and problems such as anxiety, obsessive behaviours, hoarding, paranoia, and depression have already been documented [3].

Sweden adopted early on a less restrictive strategy to combat COVID-19 based on recommendations from the Swedish Public Health Agency, emphasising individual responsibility. The elderly at retirement homes were particularly affected, along with those belonging to health-risk groups. While several occupational groups were advised to work from home, from time to time, schools for children up to 16 years were open throughout the pandemic. University students were provided education through distance learning, with the exception of some classroom teaching in August 2020. In this paper, based on a survey among university staff and students in Sweden, we focus on the resilience of Swedish academics in the face of the coronavirus.

The first part of the introduction presents some studies that are important for framing our theoretical perspective. While some international studies have shown an increase in stress, anxiety and depression among academic staff and students during the lockdown, increased by the lack of physical exercise or no recreational activity [1,4,5,6,7], others have pointed out a sense of isolation due to teleworking, as well as a lack of separation between home and workplace and a sense of flexibility and autonomy [1,8,9]. Resilience and mental health were the focus of several studies, and a US study showed that lower scores on resilience (using the Connor–Davidson Resilience Scale—CD-RISC) were associated with worse mental health outcomes (depression, suicidal ideation and severe anxiety) as well as an association between lower resilience and greater worry about the effects of COVID-19 [5]. Greater resilience, on the other hand, was predicted through social factors, such as daily outdoor activities, social support from family, friends, and close significant others, and prayer [5]. Similarly, a Spanish study showed that older adults who regularly engaged in vigorous and moderate-vigorous physical activities during quarantine reported higher scores in resilience (CD-RISC 15 items), positive effects, and lower scores in depressive symptoms [10]. In a cluster sampling study among students in China, using the Generalized Anxiety Disorder Scale (GAD-7), the majority experienced mild anxiety; it was also found that living in urban areas, family income stability, and living with parents were protective factors [1].

In other studies, resilience was related to age. Nurses working in a COVID-19 reference hospital in Iran [11] showed a moderate level of resilience (CD-RISC 25 items) that increased among older nurses with more work experience; the results also showed a significant negative correlation between resilience and hypochondriasis. A study among public workers during the pandemic in Korea concluded that high levels of stress and anxiety were defined by low resilience (CD-RISC 2), suggesting that psychological resilience may be an effective psychological intervention [6]. Based on a non-clinical Chinese population during the first wave of COVID-19, age differences in psychological resilience were found, where older participants (>55 years) showed higher resilience than the younger group (<18 years), possibly due to life experience [12]. The same study also revealed lower CD-RISC scores for the depressed group when compared to those who were not depressed. The authors recommend providing information on the risks related to COVID-19, promoting optimism, and active coping styles, as a way to mitigate the negative mental health effects of the pandemic. Gender and educational differences in resilience have also been found. An Iranian, non-clinical population study reported a high level of resilience (CD-RISC 25), more so among men and unmarried people [13]. A study on a Spanish population, investigating the resilience level (CD-RISC 10) during the pandemic, found that education had significant importance, as those with postgraduate or doctorate studies reported higher resilience as well as those who were responsible for dependents [14].

Closer to the framework of our paper, some studies have focused on coping methods among academics. Among Swedish academics, several secular existential coping methods appeared as the most common: among these, nature, followed by listening to the sounds of surrounding nature, thinking of life as part of a greater whole, walking/being active outdoors, being alone, and thinking of an internal spiritual force that exists [15]. The coping methods were similar across several subgroups. However, religious coping methods were clearly the least used methods. The presence of strong secular existential coping methods, the authors thought, could be explained by Swedish people being more open to subjective, inwardly directed spirituality rather than a religious orientation. Furthermore, being alone, which is a strong coping method among Swedes, could be best understood through the positive connotation of *ensamhet* (solitude), a quality evoking inner peace, relaxation, and personal strength, according to the authors. As part of the same research project but among Iranian academics, the most common coping method during the crisis, choosing among religious, spiritual and secular methods, was thinking that life is part of a greater whole [16]. This was mostly found among on-campus students and older age groups. The second most common coping method was praying to Allah/God or another religious figure, mostly found among the youngest age groups and women. The same population ranked the religious coping methods highly, within the top ten. Some exceptions were found among men and the age group of 30–59 years old, who expressed being alone and contemplating the crisis as the second most common coping methods. Another exception was found among people living in small towns far from a large city, who ranked listening to the sound of the surrounding nature as their top method, followed by thinking about a spiritual force, spending time alone, and thinking that life is part of a greater whole. However, in a concurrent study conducted by the present study’s research team in Iran, we found a strong level of resilience, slightly stronger among men (*M* = 5.78) than women (*M* = 5.52; no significant difference) (unpublished data). Furthermore, similar to previous results on age and gender differences, slightly more than half of the respondents had high resilience, higher for men and higher for those older than 35 years. For social and collective resilience, the data showed that almost all academics in Iran claimed, in ranking order, that their family, friends, work/school, and religion/spirituality had helped. Furthermore, among almost all respondents, general health was high, more so among women; however, men more often claimed that their health was excellent or very good. Differences across age groups were small, slightly higher among the young ones. There was a positive correlation between general health and resilience. The correlation between general health and the frequency of different coping methods revealed that the higher the usage of different coping methods, the higher the general health, and vice versa. The correlations were strongest between general health and the coping methods “life as part of a greater whole”, followed by “being alone”, “listening to the sounds of the surrounding nature”, “regularly meditating”, and “nature as an important resource”. Having family as the main coping strategy linked well to the importance of and devotion to family in Iran, the importance of the influence of the environment, and the importance of structural factors for the outcome of individual health, the authors concluded.

Though research has investigated the connection between resilience and health outcomes during the COVID-19 pandemic, limited research has been conducted on the connection between spirituality, coping skills and resilience, especially among women during the pandemic. One such mixed-method study among women [17] found in a population from the US, the majority being Christians, that higher levels of spirituality correlated positively with higher levels of resilience. By subgroups, women 65 and older and single/never married women had the highest scores in spirituality and resilience. Similarly, high resilience (CD-RISC), spirituality, and high levels of family functioning were found to be positive coping methods for nurses against stress, anxiety and depression [7].

As seen in this brief research review, most studies have approached resilience as an inner capacity, focusing on control, acceptance of change, trust in instincts, individual competency, or ability to overcome danger. Fewer studies have measured resilience as influenced by social and collective factors. Thus, the current study intends to fill this knowledge gap. This brings us to the concept of resilience itself.

### 1.2. Theoretical Framework: Personal and Social/Collective Resilience

In the second part of this introduction, we want to present the concept of resilience as a theoretical framework guiding our analysis and discussion. In the 1980s, resilience, a psychological term, was synonymously used for “the ability of individuals to recover from exposure to chronic and acute stress” [18] (p. 13). These early studies focused largely on personal features such as individuals’ ability, strength, motivation, qualities and capacities, as well as genetic predispositions, as factors prompting personal adjustment skills. Although extra-individual factors were considered, they were not centred as the focus of the research because “personal qualities” were regarded “as the sine quo non of developmental outcomes” [18] (p. 15, emphasis in original). Later studies of resilience looked more in-depth at structural factors and by focusing on “how the fabric of a society impacts individual mental health trajectories” [19] (p. 369). A culturally and contextually sensitive definition of resilience took form, which pointed at the process of navigation and negotiation, as demonstrated through the definition by the Canadian psychologist Michael Ungar [20]:


*In the context of exposure to significant adversity, whether psychological, environmental, or both, resilience is both the capacity of individuals to navigate their way to health-sustaining resources, including opportunities to experience feelings of well-being, and a condition of the individual’s family, community and culture to provide these health resources and experiences in culturally meaningful ways. (p. 225)*


Ungar’s approach emphasises the significance of social relations, approaching resilience as an interactive two-way process that is nurtured by internal perceptions as well as external stimuli [18]. This constant negotiation between the internal and external factors is, in turn, driven by opportunities and obstacles in place. Such opportunities encompass resources (social, cultural, psychological, physical), which need to be available and accessible for the individual, Ungar argues. In line with this approach to resilience, the resources must be meaningful to the individual in order to help enhance resilience, as expressed by Zittoun and Brinkmann [21] (p. 1809): “‘Meaning-making’ designates the process by which people interpret situations, events, objects, or discourses in the light of their previous knowledge and experience”.

Thus, to conclude this theoretical section, when using resilience in this article, we refer to individuals’ capacity, both intra- and interpersonal, as well as the environment’s influence on individual and collective behaviours and meanings.

### 1.3. Aims

The aim of our research is to fill the scarceness in knowledge on resilience and health among academics in Sweden who were challenged by the COVID-19 pandemic, with special attention to personal/individual and social/collective resilience. We also aim to focus on the cultural context when interpreting the results. The independent variables primarily used were gender (men/women) and age (<35/35–49/<50 years). The dependent variables were defined as personal/individual resilience (adapt/bouncing back), social/collective resilience (importance of family, religion/spirituality, nature, work/school, and friends), and general health. We also aim to test the relationship between personal/individual and social/collective resilience, general health and meaning-making coping methods. The research questions guiding this study are:Q1. How strongly do academics in Sweden rate their level of personal/individual and social/collective resilience (disaggregated by gender, age group, residence, and employment status)?Q2. How strongly do academics in Sweden rate their general health (disaggregated by gender, age group, residence, and employment status)?Q3. Is there a relationship between general health, personal/individual resilience, social/collective resilience, and meaning-making coping methods among academics in Sweden?Q4. What resilience factors contribute most strongly to meaning-making among Swedish academics, and how can this be culturally interpreted?

## 2. Materials and Methods

Prepared as a quantitative study, we developed a questionnaire (see the Appendix A) that was distributed across academic settings in Sweden.

### 2.1. Sampling

The target group consisted of staff/faculty members and students in Swedish universities and colleges. In 2020, the number of employees at Swedish universities and colleges reached 64,300 full-time equivalents (FTEs), with slightly more women than men [22]. The number of researchers and teachers was equal to 15,000 women and 17,300 men in the same year [22]. The most common employment category was senior lecturer, at one-third, followed by professors, lecturers, and other research and teaching staff without PhDs, at approximately one-sixth each, and other research and teaching staff with PhDs and career-development positions, at one-eighth [22]. Furthermore, most researchers and teaching staff, one-fourth, respectively, were found in medical and health sciences and social sciences, followed by natural sciences, engineering and technology, humanities and the arts, and agricultural sciences and veterinary medicine [22]. Among researchers and teachers, almost two-fifths had a foreign background (foreign background includes a person who is either born outside Sweden or has two parents born outside Sweden). In 2020, there were approximately 1.5 million active students at the university/college level, three years or higher [23].

Although the academic groups, staff and students were relatively homogeneous and their email addresses, to some extent, were available, for this study, we found convenience sampling most characteristic. We chose not to generalise the results from the study sample to the whole university/college and student population in Sweden. The inclusion criteria were university/college faculty/staff members and students, full- or part-time, studying any subject at Swedish universities or colleges. The exclusion criteria were not working or studying at a Swedish university/college and not being able to give informed consent.

### 2.2. Procedure

A spread of universities and colleges (*n* = 11) (covering geographical regions, larger and smaller units) from a total of 40 in Sweden were selected; from the list, we contacted research and administrative personnel (deans, director of studies, prefects and course administrators at different faculties and student health services) at the university departments. They were asked to disseminate the online survey to their staff and/or students. Data collection was done by an open online questionnaire from the University of Gävle and their web survey tool Sunet, following the handling of personal and sensitive data according to GDPR rules. Since the study was designed as convenience sampling, data collection was limited between 30 May 2020 and 1 December of the same year. The email first presented an invitation letter before participants were asked to give their consent and answer the questions. In total, 278 participants working or studying at different universities completed the questionnaire. No missing data were reported. Table 1 presents the demographics of the participants.

As seen in Table 1, the majority of participants were women. The age distribution was relatively equal, with a third in each age group: young, middle age, and older. Not surprisingly, the vast majority had a university level or equivalent education, and, similarly, a majority was born in Sweden. Among the respondents, 81% were full-time or part-time employed, and 19% were students. Fifty percent of the study population were married, followed by single and other relationships. Moreover, 64% had children, and 48% of the study sample lived in a medium-large city.

### 2.3. Measures

The survey (see the Appendix A for the formulation of the items) consisted of items linked to the theoretical framework of resilience. Resilience was gauged using two items from the Connor–Davidson Resilience Scale (CD-RISC 2): the ability to adapt to changes and bouncing back after an illness [22]. The CD-RISC was inspired by a biopsychospiritual model of adaptation to stressful events, tested on both the general population and clinical samples [24]. Respondents with a mean score of ≤2.66 are regarded as having low resilience, 2.68–5.32 moderate resilience, and 5.34–8 high resilience. The translated Swedish version was used. As the CD-RISC measures the personal/individual level of resilience, we added items on social and collective dimensions by asking about the importance of family, friends, nature, work/school, and religion/spirituality (these items were developed by Cetrez et al. [25] to measure resilience among refugees and newcomers in Sweden) during the pandemic, scored from “not at all” to “very much”. To evaluate general health, we included one item on self-perceived health, scored from poor to excellent, taken from the RAND 36-Item Health Survey 1.0 questionnaire items [26]. Furthermore, we included items on gender, age, educational level, employment, and place of residence. Some items of the Brief RCOPE instrument were used [27], while the selection of items was based on the results of other studies on meaning-making coping in Sweden [28]. To the Brief RCOPE instrument, we added, therefore, items concerning the secular existential coping methods that the research group had identified and used in different studies [28]. The modified RCOPE used here had a Cronbach’s alpha value of 0.742 (high). The instrument includes 16 items, rated on a 4-point Likert scale, ranging from 0 (“Never”) to 3 (“Always”), plus 9 background items. The instrument was validated for language and content in an earlier study [27]. Content and concepts were adjusted to fit the majority Swedish cultural context; the church, priest/pastor, and God were used.

### 2.4. Data Analysis Methods

Designing our study as convenience sampling implies that we are not doing any tests for significance between groups. Cross tabulations (by gender and age group) and correlations were performed using SPSS^®^ Statistics Version 27 (SPSS Inc., Chicago, IL USA).

## 3. Results

This section is divided into subheadings. It provides a concise and precise description of the experimental results, their interpretation, and the experimental conclusions that can be drawn.

### 3.1. Personal/Individual Resilience

The CD-RISC 2 shows the capacity to recover from a crisis such as COVID-19; the academics and staff members in Sweden were asked two questions: if they can adapt when changes occur and if they tend to bounce back after an illness, injury, or other hardships, based on a scale from 0 (not at all) to 4 (all the time). Through a cross-tabulation, personal/individual resilience was measured along with gender, age, and education. Both men (*M* = 6.05) and women (*M* = 5.90) showed a strong level of resilience. In the age category, those between 35 and 49 years old (*M* = 6.31) showed the strongest resilience, followed by those 50 years and older (*M* = 5.78) and those younger than 35 years (*M* = 5.76).

### 3.2. Social and Collective Resilience

As seen in Figure 1, on meaning-giving through social and collective resilience, almost all respondents claimed that their family and friends were somewhat (or more) helpful in making sense of life or giving life meaning during the COVID-19 crisis, followed by work/school, nature, and, lastly, religion/spirituality. While family was seen as helping very much, by a clear majority, the opposite was found for religion/spirituality, namely, not helping at all.

### 3.3. General Health

As seen in Figure 2, on general health, a majority of the respondents said their health was excellent, very good, or good, which accounts for 90%. Men more often claimed that their health was excellent; however, they also responded to a higher degree that their health was poor and fair. There were only small differences across age groups. Similar to the age differences within resilience, the age group 35–49 said that their health was excellent to a higher degree, while those 50 years or older said their health was poor, slightly more than the other age groups.

### 3.4. Religious and Spiritual Background and Thinking

As seen in Figure 3, on religious and spiritual background, half (51%) of the respondents claimed that they believed in God or another religious figure, at least somewhat. Similar figures were found for those who said they grew up in a religious family and somewhat thought that there is a higher power or benevolent power. However, a larger number (73%) responded that they tried to control a situation without the involvement of God or other religious figures.

### 3.5. Resilience, Self-Rated Health and Coping Methods

There was a weak positive correlation between personal/individual resilience and general health (*r* = 0.252). This was followed by a correlation between social/collective resilience and general health (see Table 2).

In Table 2, on the correlation between general health and meaning-giving through social/collective resilience, we see a weak positive correlation for all factors except nature.

In Table 3, on personal/individual resilience and coping methods, we see mainly weak negative correlations for religious coping methods. We performed a similar calculation for the correlation between self-rated health and coping methods, which did not correlate strongly.

## 4. Discussion

Reviewing the results of Question 1, the academic sample rates their level of personal/individual resilience across different subgroups as high, more so among men and the middle-aged. On social/collective resilience, they rate family and friends as the most meaning-giving source, while religion/spirituality is rated as the least. On general health (Question 2), the majority finds it strong, again more so among men and the middle-aged. On the correlation between general health and personal/individual resilience (Question 3), the results were weak and positive, similarly between general health and social/collective resilience. On the correlation between personal/individual resilience and coping methods, the results were mainly negative and weak. The resilience factors we can discern among the academics in Sweden that contribute to meaning-making (Question 4) are primarily family, followed by friends (although not as strongly) and nature. This is discussed in more detail in the following section.

### 4.1. Meaning-Giving Factors in Social/Collective and Personal/Individual Resilience

Earlier research has pointed out the importance of nature in Swedish culture in coping with COVID-19 [15], as well as the importance of perceiving a sacred value in nature [28], a love of nature and natural romanticism in Sweden [29], and contact with nature as being essential to one’s well-being [30]. Our study also showed the importance of nature in meaning-giving during the pandemic, although it was not as important as family and friends. Additionally, both nature and religious items correlated weakly with personal/individual resilience (CD-RISC). The importance of family and friends as resilience factors is in line with international studies [1,5,7] as well as with our theoretical approach to resilience, where family and community are important health-providing resources [20]. The preference for family, above all, may be explained by its importance in times of severe crises, offering valuable and multiple resources, such as economic resources, shared beliefs and values, affectionate rituals, traditions, support systems, and positive self-esteem [1,18], as was responsibility for dependents [31]. In other words, in times of existential crises triggered by isolation, fear of sickness and death, the need for family and social relations becomes more assertive. Furthermore, as family is linked to belief systems, organisational patterns, and communication processes [18,21,32], it also becomes meaningful and purposeful within the specific context of COVID-19. Using Ungar’s [20] definition of resilience, this indicates that academics, during the pandemic, had a good level of health and they were able to navigate their way to health-sustaining resources, which their families and culture provided in a meaningful way. In times of a pandemic, family concern, perhaps due to the fear of losing family members, seems to take priority as a meaning-making factor. This is confirmed by studies in Sweden and China showing that during the pandemic, individuals, more than before, referred to their families as other networks and activities were limited [1,33]. However, there is limited research on how the COVID-19 pandemic may have affected the role and function of family as a resilience factor. Despite the positive roles that family and community may play for resilience, we should not forget the risks of negative influence due to mal-functioning family or community conditions [34], which can intensify in times of isolation.

The results also showed that the importance of religion/spirituality for meaning-giving was very limited in times of a pandemic. So was a belief in God, a religious figure, a higher power and a religious background. Religiosity/spirituality did not show a high correlation with self-rated health either. This is understood in light of earlier studies showing that Sweden is one of the most secular countries in the world [35] and also that religious coping methods were the least used methods during the pandemic [15]. Interestingly, this differs from the positive correlation between spirituality and resilience found in the US study [17]. An explanation for the different results found for religion/spirituality and meaning-giving as well as resilience is given by cultural and socialisation theories—those for whom religion/spirituality was important before the pandemic also found it useful during the pandemic [17,36].

This is also in line with Zittoun and Brinkman’s [21] approach to meaning-making, namely, that meaningfulness needs to be evident to the individual and needs to match her/his needs, which, e.g., depends on her/his socialisation, respectively, and her/his “previous knowledge and experience” (p. 1809), thus meaning links to the system that signal to individuals the importance of certain factors within their life or, at least, in certain areas in their lives, such as well-being. Ungar [18] also argues that meaning-making guides people towards what they perceive as purposeful actions and “to which resources (opportunities) they value and access” (p. 22). Furthermore, Ungar’s argument is that the resources provided depend on the meaning that is attributed to them, usually indicated by the dominant culture within a specific socio-cultural, socio-historical and time-specific framework.

### 4.2. Resilience and Health

Previous research shows, with limited exceptions, a positive correlation between resilience, physical as well as mental health, and religiosity and spirituality. The results are not surprising, as there is an expected positive correlation between resilience and health, also seen in many other studies [5,6,7,10,11,31]; however, they did not confirm positive correlations between health and religion/spirituality [7,17]. Still, there may be explanatory factors such as a healthy childhood, socioeconomic, physical and mental conditions, being an active person, and a stronger network, among others. When looking at subgroups, our results also confirmed earlier research, where a stronger resilience level was found among older age groups [6,11,17] and among men [13].

## 5. Conclusions

Resilience, both on personal/individual and social/collective levels, is important for general health during times of crises such as COVID-19, with some interesting subgroup differences. Often overlooked but important in order to understand this outcome are two aspects, the time dimension and context. Researchers point to post-trauma growth, where individuals can learn and recover from adversity [31]. Thus, firstly, this explains why we find higher resilience among older age groups, as this is a group that has built up resilience by experience and age. Second, central to any understanding of resilience and health outcomes is the cultural context. Starting from the definition of culture provided by the cultural psychologist Anthony Marsella [37], who refers to culture as “shared learned behaviour and meanings that are socially transferred in various life activity settings for purposes of individual and collective adjustment and adaptation” (p. 657), we may conclude that the context for resilience is both dynamic and enduring, i.e., practices promoting resilience are both subject to change as well as static for some time. Furthermore, again linking to the definition of culture by Marsella, resilience is shaped by our cultural worldview, perceptions and orientations in life, as well as our concepts of normality/abnormality. Thus, despite the fact that Sweden is a strong individual-oriented society, the fear of losing family members due to the high number of deaths in COVID-19 evoked emotions related to family, which reflects a different cultural orientation during a specific time period, such as the pandemic. However, a contextual explanation may be that academics as a sub-population are highly resilient. Their resilience is assumed to be linked to their qualities of flexibility, adaptability, and emotional resilience, being collaborative, empathetic and open-minded, as well as their willingness to demonstrate collective responsibility and their older age level and better economic situation (in general), reducing their risk of stress [9]. The same authors have also pointed out the multidimensional and multi-levels of resilience, linking them to the bioecological system approach, which acknowledges the individual nature of resilience and places this within the broader ecological context, i.e., the micro, meso, and macro levels.

Earlier research has shown noticeable geographical and cultural value differences that may explain the different and, at times, contradictory results on resilience and health during COVID-19. These cultural differences can be understood in light of the World Values Survey results. In some contexts, traditional and survival values are strong (Iran), and traditional values and self-expression values are moderate (Spain, US). In others, secular values are strong and self-expression values are moderate (China, Korea), and in yet others, secular and self-expression values are very strong (Sweden) [38]. Thus, context and culture need to be part of the interpretation in order to understand the differences found in these geographically widespread studies.

## 6. Limitations

One limitation of the current study is the convenience sampling frame, with relatively small subgroups, making the level of representativeness and generalisation limited. Additionally, some subgroups, such as men, individuals of older age, divorced and engaged persons, those having children, and those living in a small town far from a large city, were underrepresented. Second, while the CD-RISC 2 is a good instrument for measuring the individual dimension of resilience, a validated instrument measuring the social and collective dimensions of resilience would be an important contribution to research. Statistical studies are useful for generalising; however, to capture the experiences and processes of the meaning-making dimensions of resilience, in-depth interviews would be more appropriate and informative. Finally, we reached out specifically to academics; thus, our results do not reflect other employed categories at universities, such as cleaners and janitors. Despite these limitations, our study is novel and may contribute to building our knowledge base on resilience in times of crisis.

## 7. Future Research and Policy Recommendations

As our review has revealed and as confirmed by an earlier review [9], few studies have focused on the resilience of the academic community during COVID-19. However, this may change in the future as COVID-19, compared to earlier pandemics, has meant a substantial disruption of the educational system worldwide [39]. Based on the results of this study, we suggest a few specific recommendations that are relevant to policy or practice:Focus on the bioecological framework of resilience in the context of COVID-19 by paying attention to individual, environmental, and meaning dimensions among academic staff.This study conducted a simple analysis univariate correlation (Pearson’s correlation). A multivariate analysis, such as regression analysis, adjusting for age, sex, and location, is required for future research to strengthen our findings.Set up structural resources in society to better balance individual capacity resources for resilience. Here, we refer to social resources such as social networks, stronger and wider connections to meaningful groups such as families and workmates, and also engaging in activities related to university or other groups, such as neighbourhood groups.Give due credit to academic staff as first responders within the education sector during pandemics as they show success in fulfilling their work duties without taking away the responsibility of their institutions to provide the necessary resources.

## Figures and Tables

**Figure 1 ijerph-19-13346-f001:**
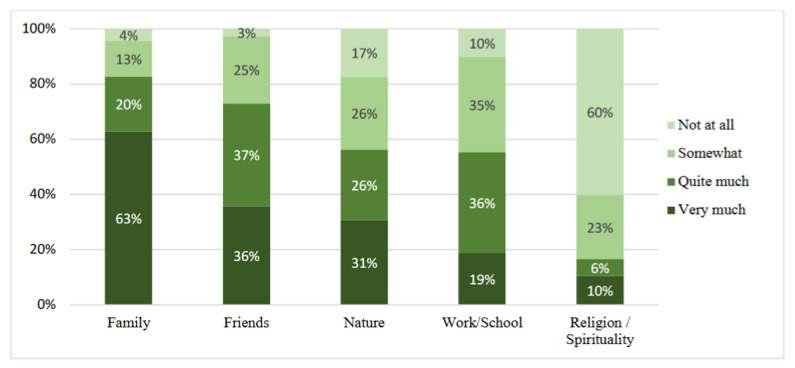
Meaning-giving and social/collective resilience during COVID-19, by percentage.

**Figure 2 ijerph-19-13346-f002:**
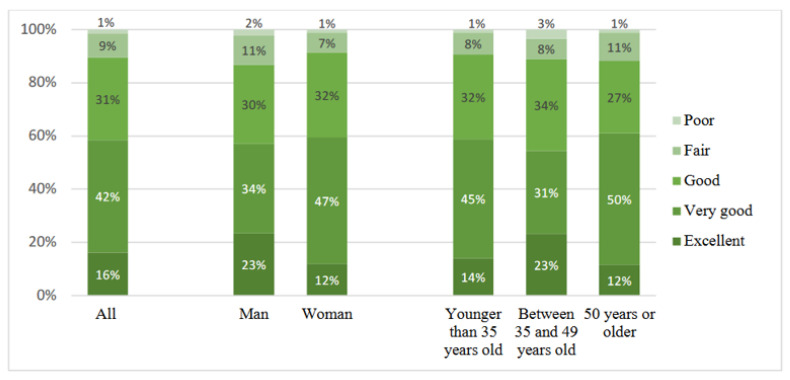
General health during COVID-19 by gender and age group, by percentage.

**Figure 3 ijerph-19-13346-f003:**
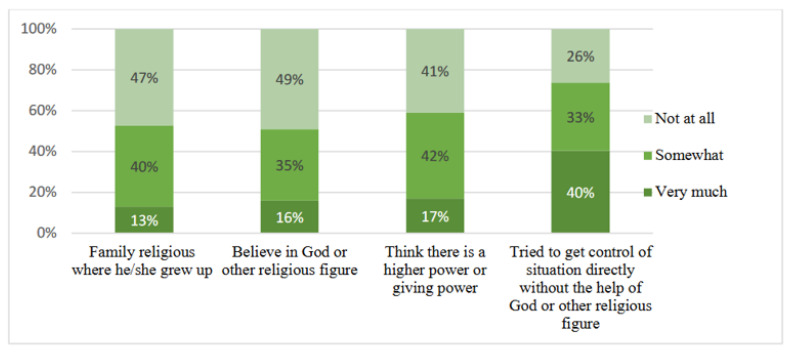
Religious and spiritual background and thinking, by percentage.

**Table 1 ijerph-19-13346-t001:** Sample characteristics (*n* = 278), by percentage.

Variable	Variable Value	%
Gender	Man	36
	Woman	64
Age groups	Younger than 35 years old	31
	Between 35–49 years old	32
	50 years or older	37
Education	University	94
	Gymnasium or similar	6
Country of birth	Sweden	75
	Other countries	25
Job/student situation	Full-time employment	66
	Part-time employment	15
	Student	19
Social status	Married	50
	Divorced	5
	Engaged	11
	Widowed	1
	Single	17
	Other	16
Children	Having children	64
	No children	36
Place of living	Capital	18
	Medium-large city	48
	Small town	34

**Table 2 ijerph-19-13346-t002:** Correlation between meaning-giving through social/collective resilience factors and general health.

		Family	Friends	Religion/ Spirituality	Work/ School	Nature
General health	Pearson Correlation	0.027	0.044	0.045	0.048	−0.020
	*n*	278	278	278	278	278

**Table 3 ijerph-19-13346-t003:** Correlation between personal/individual resilience and frequency of different coping methods.

		P1	P2	P3	P4	P5	P6	P7	P8	P9	P10	P11	P12	P13	P14	P15	P16
CD-RISC	Pearson Correlation	−0.029	−0.009	−0.059	0.081	0.038	−0.049	0.010	−0.189	−0.157	−0.167	−0.026	−0.239	−0.062	−0.113	−0.119	0.050
	*n*	278	278	278	278	278	278	278	278	278	278	278	278	278	278	278	278

Note: Dark green shows the strongest correlation and dark red the weakest correlation. P1. Have you thought that your life is part of a greater whole? P2. Have you thought or felt that a spiritual force exists in you to help you deal with the situation? P3. Has nature been an important resource for you in how to deal with your stress/sadness or other negative feelings? P4. Has being alone and having the chance to contemplate help you deal with the situation? P5. Have you listened to the sounds of the surrounding nature? P6. Have you walked or engaged in any activities outdoors that give/gave you a spiritual feeling? P7. Have you regularly meditated when dealing with your stress/sadness or other negative feelings? P8. Have you sought spiritual help from a religious leader? P9. Have you thought that COVID-19 was caused by an evil power? P10. Have you wondered if God has left you or become angry that God is not present to help you? P11. Have you had the feeling of a strong connection with God? P12. Have you visited the church, synagogue, mosque, temple, or other religious places? P13. Have you prayed to God or other religious figures to make things better? P14. Have you listened to religious or spiritual music? P15. Do/did you think that you have done your best and now it is only God who is in control? P16. Have you tried to get control of your situation directly without the help of God or other religious figures?

## Data Availability

The data presented in this study are available upon request from the corresponding author. The data are not publicly available due to privacy.

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
