# Peer review of "When a Pandemic Strikes: Resilience of Swedish Academics in the Face of Coronavirus"

_ijerph, 2022, doi:10.3390/ijerph192013346_

Round 1

Reviewer 1 Report

Dear authors, thank you for the opportunity to review your paper, and congratulations on your work.

The manuscript deals with a very interesting topic such as the study of resilience in Swedish university staff during the COVID-19 pandemic.

The paper is well organized and structured. The objectives and questions set out are clearly defined and have been resolved one by one throughout the work. All the data necessary to justify the conclusions reached are presented.

However, I find some shortcomings that need to be explained or corrected, as they are adapted to the study you have carried out and not to a possible new scenario, if necessary.

Below is a list of the manuscript:

The abstract is well structured, with well-defined parts and its length is correct.

The keywords are representative of the paper, although they should include some more specific or even mesh terms.

The introduction section is well laid out, going from a more general character to a more specific one. Interesting topics such as the different papers are presented.

The Material and method section is very well explained, with great detail on the population to which the study refers and the probability sampling to achieve a homogeneous sample. Congratulations, it is difficult to see such detail in an article.

When we refer to software we usually add the responsible company behind it, e.g. IBM® SPSS® Statistics Version 26 (Ar-153 monk, NY, US).

Question 1: Is the sample (n=278) representative? Calculating, we need an n=384 (with a confidence size of 95%), which leads us to be unable to generalize the results. Furthermore, they explain in their limitations that there is undersampling in some categories.

Regarding the statistical analysis, there is much controversy about whether Likert scales should be considered quantitative (interval) or qualitative (ordinal) variables and this influences the choice of statistical tests to be carried out, parametric or non-parametric. In my opinión, I am  referring to qualitative variables, but many references in the academic world consider the opposite. I find that they use non-parametric tests (Chi-square and contingency table) with parametric (Pearson correlation), I understand that assuming that the variables are continuous (interval) based on the existing literature.

To do so, they are based on the fact that they must have two criteria to be considered interval variables: the existence of zero and the equidistance of their options. Both are fulfilled in part of their questionnaire (the questions with five answers, items 26 and 27) although the neutral question must be the central one, so that it is equidistant between negative and positive perceptions.) The rest of the variables should be considered ordinal and therefore use non-parametric tests (t Pearson's correlation coefficient is not).

Question 2: Can you justify the use of these variables as quantitative variables for use with parametric tests?

Question 3: If the use of parametric tests persists, what test have you used to justify the assumptions of normality?

Question 4: Has the association between the variables been verified before correlation?

Line 236 notes that they talk about measuring mental and physical health (item 27).

Question 5: The question talks about health, without differentiating between mental and physical health. Even in sub-section 3.2. Self-rated health, only self-rated health is mentioned, should the proposed wording be changed?

The results obtained are presented appropriately, they are comprehensible, useful, and significant for further research in the field.

It is normal in scientific writing to refer to statistical significance (SPSS concept) as a p-value.

The conclusions are written clearly and succinctly, coherent with the objectives set and the results obtained.

In the references section, there are self-citations, from 11 to 13, which are linked to other articles in which some of the authors of the study have participated. In addition, reference 13 is under review, so it is not appropriate for citation. I understand that they are experts in their field and should use their material, but this is a large number of self-references.

Also, the numbering bullets are double-numbered.

No incidents in other sections.

Please do not understand my suggestions as criticism of your study, on the contrary, I consider it very complete and I hope you can publish it with those doubts. Looking forward to your replies.

Regards.

Author Response

Dear Reviewer,

Thank you very much for your valuable comments and suggestions, which help us improve the manuscript in different ways. Please, see our answers to your points as follows:

Reviewer (R): The keywords are representative of the paper, although they should include some more specific or even mesh terms.

  • Answer (A): We agree and suggest to add additional keywords: university staff, higher education, crisis, epidemic

R: When we refer to software we usually add the responsible company behind it, e.g. IBM® SPSS® Statistics Version 26 (Ar-153 monk, NY, US).

  • A: It is updated to: SPSS® Statistics Version 27

Question 1: Is the sample (n=278) representative? Calculating, we need an n=384 (with a confidence size of 95%), which leads us to be unable to generalise the results. Furthermore, they explain in their limitations that there is undersampling in some categories.

  • A: As the academic groups, staff and students, were relatively homogeneous and e-mail addresses to some extent were available, our aim initially was to perform a simple random sampling. We subcontracted the task, but looking at the results, we saw the sampling was not satisfactory, due to different reasons in obtaining a satisfactory register of the population and in the procedure in how it was performed. We concluded the obtained response to be skewed towards an overrepresentation of academic staff and underrepresentation of students. A subsample of 223 were employed at the universities and 54 were students, wereof 36 were campus students and 18 distance learning students. This means that 81 percent are employees and 19 percent students in the sample. There is a clear over representativeness for employees compared to students. Thus, the sampling limitations, having difficulties obtaining probability sampling, forced us to reframe our study as a convenience sample. Therefore, our aim is not to generalise the results from the study sample to the whole population; i.e., all university/college staff and students in Sweden. This also implies we are not doing any tests for significance on the results. We have added a sentence to the limitations concerning this point (Section 2.1. Sampling and section 6. Limitations). Despite the convenience sampling, we obtained both similarities and differences in results, compared to international studies; which is reflected in the Discussion.

R: Regarding the statistical analysis, there is much controversy about whether Likert scales should be considered quantitative (interval) or qualitative (ordinal) variables and this influences the choice of statistical tests to be carried out, parametric or non-parametric. In my opinión, I am  referring to qualitative variables, but many references in the academic world consider the opposite. I find that they use non-parametric tests (Chi-square and contingency table) with parametric (Pearson correlation), I understand that assuming that the variables are continuous (interval) based on the existing literature.

  • A: Indeed, as the reviewer rightly points out, there is a controversy in the literature. The variables used for the coping methods were ordinal scale using the variable values of; Never – 0, Sometimes -1, Quite often -2, Very often – 3, respectively how often they cope with the challenges when working from home; Never-1, Seldom-2, Sometimes-3, Often-4 and Always-5. But, as indicated above, to the first question, we choose to not perform any significant tests. However, still we can state that there are differences in the sample and thereafter resonate on possible causes, which we are trying to do in the Discussion.

R: To do so, they are based on the fact that they must have two criteria to be considered interval variables: the existence of zero and the equidistance of their options. Both are fulfilled in part of their questionnaire (the questions with five answers, items 26 and 27) although the neutral question must be the central one, so that it is equidistant between negative and positive perceptions.) The rest of the variables should be considered ordinal and therefore use non-parametric tests (t Pearson's correlation coefficient is not).

  • A: Spearman would be the most fitting correlation analysis to use, since we have ordinal scale instead of numeric/interval scale. Since some statisticians state that the outcome is very much the same, we stay with the results presented from Pearson’s correlation analysis.

Question 2: Can you justify the use of these variables as quantitative variables for use with parametric tests?

  • A: Please, see the discussion above, as we are not performing any parametric tests.

Question 3: If the use of parametric tests persists, what test have you used to justify the assumptions of normality?

  • A: Please, see above.

Question 4: Has the association between the variables been verified before correlation?

  • A: Yes, a reliability test was conducted and the Cronbach Alpha value was above the recommended value of 0,7, namely 0,742. This has been added to the manuscript, page 7, in Measures.

Question 5: The question talks about health, without differentiating between mental and physical health. Even in sub-section 3.2. Self-rated health, only self-rated health is mentioned, should the proposed wording be changed?

  • A: This item measures general health, taken from the RAND 36-Item Health Survey 1.0 Questionnaire Items; available here: https://www.rand.org/health-care/surveys_tools/mos/36-item-short-form/survey-instrument.html. This is now updated in section 2.3 and section 3.3 (earlier 3.2) 

R: It is normal in scientific writing to refer to statistical significance (SPSS concept) as a p-value.

  • A: As discussed above, we chose not to conduct significance tests.

R: In the references section, there are self-citations, from 11 to 13, which are linked to other articles in which some of the authors of the study have participated. In addition, reference 13 is under review, so it is not appropriate for citation. I understand that they are experts in their field and should use their material, but this is a large number of self-references.

  • A: We have deleted one of the three self-citations, nr 13, as it is still unpublished. Consequently, the numbering of references has been updated. In text it is referred to as unpublished, while it is deleted from the reference list. If possible, we wish to keep the other two, which are relevant and necessary.

R: Also, the numbering bullets are double-numbered.

  • A: These have been corrected.

Reviewer 2 Report

This is a clear and concise article that makes a contribution to the literature. With its focus on the pandemic, it is very timely.

The literature review is very strong.

The sample size of 278 is strong. The instruments utilized are appropriate.

I am not a methodologist/statistician and I hope another reviewer can be helpful here.

A strength of this work is the expansiveness of the definition of resilience.

Overall, we learn more about the heavy reliance upon and concern for family members during the pandemic. And somewhat surprisingly, less importance placed upon religion and spirituality.

I have just a few comments that I hope are helpful to the authors.

In line #84, the word "indolence" is used among "inner peace" and "personal strength." I'm not sure the word "indolence" is the correct word--this may be an issue of translation, I can't tell. In the U.S. "indolence" has a negative connotation and is defined as "laziness." The authors could consider "relaxation" as a substitute.

Figure 1 uses a scale of "not at all, "somewhat,""quite much" and "very much." The problem is distinguishing between the "quite" and "very" much categories. In the U.S. these are not easily distinguishable categories.

Line #252, in the first sentence "may be divided by" should be replaced with "is divided into." Second sentence, "should provide" should be replaced by "It provides a concise...."

The Discussion section is very strong as is the Conclusions section. In the Further Research section, I recommend the authors add another sentence or two to the second bullet that can give readers a better sense of how to revise and adapt structural resources in meaningful ways.

Overall, a very good effort.

Author Response

Dear Reviewer

Thank you very much for your valuable comments and suggestions, which help us improve the manuscript in different ways.

Reviewer (R): In line #84, the word "indolence" is used among "inner peace" and "personal strength." I'm not sure the word "indolence" is the correct word--this may be an issue of translation, I can't tell. In the U.S. "indolence" has a negative connotation and is defined as "laziness." The authors could consider "relaxation" as a substitute.

  • Answer (A): Thank you, we have replaced term “indolence” with “relaxation”.

R: Figure 1 uses a scale of "not at all, "somewhat," "quite much" and "very much." The problem is distinguishing between the "quite" and "very" much categories. In the U.S. these are not easily distinguishable categories.

  • A: Thank you for pointing this out. The translation of the Swedish term “ganska” has been suggested to be “quite”. We could also translate it to “pretty”, “fairly” or “rather”, choosing to stay with “quite”.

R: Line #252, in the first sentence "may be divided by" should be replaced with "is divided into." Second sentence, "should provide" should be replaced by "It provides a concise...."

  • A: Thank you, we have changed accordingly.

R: In the Further Research section, I recommend the authors add another sentence or two to the second bullet that can give readers a better sense of how to revise and adapt structural resources in meaningful ways.

  • A: We have added the following to the second bullet: Here we refer to social resources such as social networks, stronger and wider connections to meaningful groups like families and workmates, and also engaging in activities related to university or other groups, like neighbourhood.

Reviewer 3 Report

I appreciate the opportunity to review the manuscript #ijerph-1924094 “When a Pandemic Strikes: Resilience of Swedish Academics in the Face of Coronavirus”.

“This study aimed to understand the individual and collective dimensions of among academics in Sweden during the early wave of the pandemic [...] We used simple random sampling, administered through an online survey, among academics at Swedish universities (n=278 , 64% women). We employed the CD-RISC 2 (The Connor-Davidson Resilience Scale) to measure personal/individual resilience, additional items for social/collective resilience, and a meaning-making coping instrument (Meaning, Control, Comfort/Spirituality, Intimacy/Spirituality, Life Transformation).”

It is a study with citation potential, with a limited methodology whose results are significant for nursing, especially for public health. The Strengthening the Reporting of Observational Studies in Epidemiology (STROBE) protocol was used to evaluate this manuscript.

In the introduction, the authors present an important overview of the measures to contain the pandemic in the data collection country, whose strategies were a little different in the rest of the world, especially due to the restricted period of social isolation. In addition, they present the relationship between common mental disorders and resilience, focusing on the university community. They also present the Connor-Davidson Resilience Scale - CD-RISC and the distribution of assessment scores, together with international studies that show its application.

The objective is delimited and the research questions are consistent with the objective.

The method is limited, but does not meet all STROBE items, so it is necessary to review the following indications:

1. Absence of study design in the title and abstract;

2. Absence of a theoretical framework;

3. Absence of pre-existing hypotheses;

4. Absence of eligibility criteria and participant selection methods;

5. Absence of presentation of confounding factors;

6. Absence of explanations about sample size;

The review of the research protocol by a Research Ethics Committee was cited, thus respecting the international provisions in force.

The presentation of the results is clear, being carried out with the support of valid descriptive statistical measures. The presentation of graphs and tables does not generate doubts about the results. I suggest the presentation of the SPSS software version. The discussion is impoverished and can be expanded with the use of similar international studies.

I also emphasize the fact that 25% of the references are more than 5 years old since their publication.

Aware of the quality of this research, I indicate my assent to the publication after the aforementioned modifications.

Author Response

Dear Reviewer

Thank you very much for your valuable comments and suggestions, which help us improve the manuscript in different ways.

  1. Absence of study design in the title and abstract;
  • Answer (A): We prefer not to change the title. However, we added to the abstract the following sentence: “A quantitative research design was applied for this cross-sectional study.”
  1. Absence of a theoretical framework;
  • A: Our Introduction, first section, is an overview of previous research. The second section, however, 1.1. Personal and Social/Collective Resilience, is more specifically presenting the theoretical framework that guides interpretation and discussion. This has now been clarified by a new subheading in Introduction and a sentence in section 1.2. As follows: “In the second part of the introduction, we want to present the concept of resilience as a theoretical framework guiding our analysis and discussion.”
  1. Absence of pre-existing hypotheses;
  • A: Some of our research questions are descriptive. Additionally, the theoretical framework and previous research on resilience, in the context of coping methods among academics is limited, thus, not providing sufficient background for formulating any hypotheses. 
  1. Absence of eligibility criteria and participant selection methods;
  • A: In sections 2.1 Sampling and 2.2. Procedure, we have detailed the sampling characteristics, the sampling procedure, the spread of universities/colleges, how the online survey was disseminated, the data collection and the open online questionnaire, as well as the information letter and consent request. We have updated this section slightly, to meet the reviewers comment, adding information on exclusion criteria as well as the handling of data.
  1. Absence of presentation of confounding factors;
  • A: As the academic groups, staff and students, were relatively homogeneous and e-mail addresses to some extent were available, our aim initially was to perform a simple random sampling. However, there is a clear over-representativeness for employees compared to students. Thus, the sampling limitations, having difficulties obtaining probability sampling, forced us to reframe our study as a convenience sample. Therefore, our aim is not to generalise the results from the study sample to the whole population; i.e., all university/college staff and students in Sweden. This also implies we are not doing any tests for significance on the results. We have added a sentence to the limitations concerning this point (Section 2.1. Sampling and section 6. Limitations). Despite the convenience sampling, we obtained both similarities and differences in results, compared to international studies; which is reflected in the Discussion. We have added a sentence to the limitations concerning this point (Section 2.1. Sampling and section 6. Limitations). Neither do we make any conclusions on the causal relationships between variables. Accordingly, we have adjusted the result section 3.5 on the correlation between personal/individual resilience and self-rated health, as well as between meaning giving through social/collective resilience factors and self-rated health. In the Discussion we resonate on the confounding factors. The results are not surprising, as there is an expected positive correlation between resilience and health, also seen in other studies. Still, there may be explanatory factors, such as a healthy childhood, socioeconomic and mental conditions, being an active person, a stronger network. 
  1. Absence of explanations about sample size;
  • A: As above, designing the study as a convenience sample, we limited the data collection to a specific time period, until we achieved a minimum number of responses. This is clarified in section 2.2. Procedure. We would have prefered a larger sample, but had to accept the results we received.

7. I suggest the presentation of the SPSS software version

  • A: Indeed, we have specified it as follows: SPSS® Statistics Version 27.

8. The discussion is impoverished and can be expanded with the use of similar international studies.

  • A: We have expanded both the research review and the discussion with similar studies. Please, see references 1-3.

9. I also emphasize the fact that 25% of the references are more than 5 years old since their publication.

  • A: The older references (7 in number) are mainly related to the concepts of resilience, coping, and culture or to method, which is expected. Others are related to the instruments, which also is understandable. The empirical references on COVID-19 are however new and up to date.

Reviewer 4 Report

The present study aims to contribute to the increased understanding of the effects of COVID-19 pandemic on the health of academics. The topic is very interesting and very important for building strategies that promote productivity and well-being in education institutions.

The title is appropriate and indicates the main message of the paper and the abstract is well structured and clear. The objectives of the study are clearly and explicitly defined at the end of the introduction. The article provides a good and generalized background of the topic through a literature review with recent articles.

Overall, the work is very well structured and the results are clearly presented and discussed.

The limitations are well identified and future lines of research are pointed out.

However, some changes/clarifications are suggested:

1 - The sentence “While some international studies showed an increase of stress, anxiety and depression among students during the lockdown, increased by the lack of physical exercise or no recreational activity” should be better supported. A single reference is not sufficient to support this statement.

The same observation for “others pointed out a sense of isolation due to teleworking, as well as a lack of separation between home and workplace, but also a sense of flexibility and autonomy”.

2 - What does “healtMh” mean?

3 - Which version of the CD-RISC was used in this study? And why was that the version chosen?

4 - The scale developed by Cetrez et al. is the same as the one they call Brief RCOPE? This explanation is confusing and it is not clear how the scale is constructed to measure the the social and collective dimensions.

5 – “Both men (M=6.05) and women 259 (M=5.90) showed a strong level of resilience, with no significant difference. By age category those between 35 and 49 years old (M=6.31) showed the strongest resilience, followed by 50 years and older (M=5.78) and those younger than 35 years (M=5.76).” Are the differences found in the age category significant? What is the significance value? And in the other categories are there no differences? For example in education, country of birth, etc.

6 – And about the Self-rated health? Are the differences found significant? The same observation for Religious and spiritual background and thinking.

7 - The text of section 3.4 is confusingly written, it should be revised.

8 - The text of paragraph 3.4 is confusingly written, it should be revised. Moreover, the table does not show any significant correlation (**).

9 – What does a separate calculation mean? “ In a separate calculation, self-rated health didn’t correlate strongly, nor significantly with any of the coping methods.”

Author Response

Dear Reviewer

Thank you very much for your valuable comments and suggestions, which help us improve the manuscript in different ways.

1 - The sentence “While some international studies showed an increase of stress, anxiety and depression among students during the lockdown, increased by the lack of physical exercise or no recreational activity” should be better supported. A single reference is not sufficient to support this statement.

  • Answer (A): These parts have been supported with additional references.

The same observation for “others pointed out a sense of isolation due to teleworking, as well as a lack of separation between home and workplace, but also a sense of flexibility and autonomy”.

  • A: As above, this has been supported with additional references.

2 - What does “healtMh” mean?

  • A: This was a spelling mistake. We have corrected to “health”:

3 - Which version of the CD-RISC was used in this study? And why was that the version chosen?

  • A: We used the validated CD-RISC 2 item, which is mentioned in section 2.3; that is the shortest of the three CD-RISC instruments. The reason was based on our attempt not to expose the participants to a longer survey than necessary. 

4 - The scale developed by Cetrez et al. is the same as the one they call Brief RCOPE? This explanation is confusing and it is not clear how the scale is constructed to measure the social and collective dimensions.

  • A: We are not using the same instrument, the Brief RCOPE, and this is not linked to social and collective dimensions, rather to secular and existential coping. This is now explained, hopefully, more clearly in the text, as follows: “Some items of the Brief RCOPE instrument were used [27], while the selection of items was based on the results of other studies on meaning-making coping in Sweden [28]. To the RCOPE instrument we added, therefore, items concerning the secular existential coping methods which the research group has identified in different studies [28].” 

5 – “Both men (M=6.05) and women 259 (M=5.90) showed a strong level of resilience, with no significant difference. By age category those between 35 and 49 years old (M=6.31) showed the strongest resilience, followed by 50 years and older (M=5.78) and those younger than 35 years (M=5.76).” Are the differences found in the age category significant? What is the significance value? And in the other categories are there no differences? For example in education, country of birth, etc.

  • A: As the academic groups, staff and students, were relatively homogeneous and e-mail addresses to some extent were available, our aim initially was to perform a simple random sampling. However, there is a clear over-representativeness for employees compared to students. Thus, the sampling limitations, having difficulties obtaining probability sampling, forced us to reframe our study as a convenience sample. Therefore, our aim is not to generalise the results from the study sample to the whole population; i.e., all university/college staff and students in Sweden. This also implies we are not doing any tests for significance on the results. We have added a sentence to the limitations concerning this point (Section 2.1. Sampling and section 6. Limitations). Despite the convenience sampling, we obtained results both similar and different to international studies; which is reflected in the Discussion. The results obtained for gender and resilience are the mean values for the sample and not the population. See section 3.1.

6 – And about the Self-rated health? Are the differences found significant? The same observation for Religious and spiritual background and thinking.

  • A: As above, our ambition was not to perform significance tests; see section 3.3.

7 - The text of section 3.4 is confusingly written, it should be revised.

  • A: It is now re-written, hopefully, easier to understand.

8 - Moreover, the table does not show any significant correlation (**).

  • A: That is correct, as we didn’t obtain any significant differences we have deleted the note.

9 – What does a separate calculation mean? “ In a separate calculation, self-rated health didn’t correlate strongly, nor significantly with any of the coping methods.”

  • A: We clarified the sentence as follows: “We performed a similar calculation for the correlation between self-rated health and coping methods, which didn't reveal any significant differences, nor correlate strongly.”

Round 2

Reviewer 1 Report

Thanks to the authors for their modifications and explanations of the original article. I consider the current manuscript to be more complete than the previous one.

I understand your explanation about the Likert variables and the use of Pearson's correlation coefficient (I still recommend the use of Spearman's correlation coefficient as it is the most adapted to your data).

The only addition that should be made in this document is in section 2.4 (Data analysis methods), where it is specified that only cross-tabulations have been carried out, however, in section 3.5 (Resilience, self-rated health, and coping methods) the use of Pearson's correlation coefficient (r) is emphasized. This has to be stated in Data analysis methods.

And then one more cosmetic correction, to make it more readable, in table 3. Instead of the complete question, I would number them (e.g. P1, P2...) and detail these in the text.

Congratulations on the work done.

Regards

Author Response

Dear Reviewer,

Thank you for your feedback and suggestions. We have updated section 2.4, adding correlation analyses. We have also changed Table 3, with notes for each item, as you suggested. 

Thank you very much for your time and valuable comments.